# Antioxidant Effects of Dietary Supplements on Adult COVID-19 Patients: Why Do We Not Also Use Them in Children?

**DOI:** 10.3390/antiox11091638

**Published:** 2022-08-24

**Authors:** Veronica Notarbartolo, Claudio Montante, Giuliana Ferrante, Mario Giuffrè

**Affiliations:** 1Department of Health Promotion, Mother and Child Care, Internal Medicine and Medical Specialities, University of Palermo, 90128 Palermo, Italy; 2Department of Surgical Sciences, Dentistry, Gynecology and Pediatrics, Pediatric Division, University of Verona, 37134 Verona, Italy

**Keywords:** antioxidants, children, SARS-CoV-2, COVID-19, respiratory viral infections, prophylaxis, therapy, oxidative stress

## Abstract

Respiratory tract infections (RTIs) are very common in children, especially in the first five years of life, and several viruses, such as the influenza virus, Respiratory Syncytial Virus, and Rhinovirus, are triggers for symptoms that usually affect the upper airways. It has been known that during respiratory viral infections, a condition of oxidative stress (OS) occurs, and many studies have suggested the potential use of antioxidants as complementary components in prophylaxis and/or therapy of respiratory viral infections. Preliminary data have demonstrated that antioxidants may also interfere with the new coronavirus 2’s entry and replication in human cells, and that they have a role in the downregulation of several pathogenetic mechanisms involved in disease severity. Starting from preclinical data, the aim of this narrative review is to evaluate the current evidence about the main antioxidants that are potentially useful for preventing and treating Severe Acute Respiratory Syndrome Coronavirus 2 (SARS-CoV-2) infection in adults and to speculate on their possible use in children by exploring the most relevant issues affecting their use in clinical practice, as well as the associated evidence gaps and research limitations.

## 1. Introduction

Respiratory tract infections (RTIs) are very common in children, especially in the first five years of life, and several viruses, such as the influenza virus, Respiratory Syncytial Virus (RSV), and Rhinovirus, are triggers for symptoms that usually affect the upper airways. Although being a benign and self-limiting condition, RTIs have a significant medical burden and social cost [1]. It has been known that during respiratory viral infections, a condition of oxidative stress (OS), due to a pro-oxidant–antioxidant imbalance, occurs [2,3] with a reduction in the cellular antioxidant defense system [4]. Two are the main reactive species produced: reactive oxygen species (ROS) and reactive nitrogen species (RNS), both of them associated with an increased risk of tissue injury [3]. Increased production of ROS, secondary to respiratory viral infections, has been reported to contribute to airways’ inflammation and tissue damage [4]. Evidence suggests that during respiratory viral infections, the use of antioxidants may help reduce symptoms and facilitate recovery in affected children [5,6], especially if they are administered early before the beginning of a severe pro-inflammatory response in the host [7]. For instance, preclinical studies demonstrated that ascorbic acid is an essential factor for the antiviral immune responses against influenza A virus (H_3_N_2_) in the early stages of the infection [8]. At the same time, Huang et al. have shown, in vitro, the suppressive role of melatonin on RSV infection by modulating toll-like receptor (TLR)-3 signaling [9]. Additionally, zinc supplementation would be fundamental in decreasing OS as well as in shortening the duration of symptoms in adults, due to its direct antiviral effect on RSV, dengue virus and coronaviruses [5]. Starting from these assumptions, many researchers have focused their attention on the role that antioxidants could have as complementary components in prophylaxis and/or therapy of respiratory viral infections in adults and children [10]. In fact, antioxidants are able to elicit immune-boosting properties [10] by increasing the number of circulating white cells and reducing ROS level [11]. Furthermore, antioxidant deficiency seems to be correlated to an alteration in T-cells’ activity and a dysregulation of the antibody-mediated immune response, increasing an individual’s susceptibility to infections [12]. Antioxidants can be divided into three groups: those that enhance endogenous antioxidant enzymes, with acceleration of inactivation of free radicals (i.e., N-Acetyl-Cysteine, that reduces the formation of proinflammatory cytokines), non-enzymatic scavengers (dietary antioxidants) and other compounds affecting OS (i.e., using oxygen supplementation and inhaled corticosteroids together) [13]. A large proportion of endogenous antioxidants are encoded by the Nuclear Erythroid Factor 2-Related Factor 2 (Nrf2) pathway; alternatively, exogenous antioxidants are derived from the oral intake of synthetic formulations or a normal diet [14]. Nrf2 is a transcription factor that contributes to the expression of cell-protective genes in response to OS; in particular, it is able to activate several antioxidant genes such as superoxide dismutase 1 (SOD1) and glutathione (GSH) [15]. High intracellular concentrations of glutathione are required to support several redox reactions [16]. Transcription of Nrf2 is inhibited: under OS, Nrf2 translocates into the nucleus and activates an antioxidant response in the elements battery of genes [17]. Recently, Transient Receptor Potential (TRP) channels have also been studied for their important role in mediating airway tissue injury and inflammation. Among them, Vanilloid-TRP 1 (TRPV1) works as a multisensory receptor for damage signals and plays an important role in the transduction of noxious stimuli and in the maintenance of inflammation [18]. Further, the TRP-ankyrin 1 (TRPA1), expressed by nociceptors, is activated by oxidizing substances and plays a major role in modulating inflammatory pain [19]. Furthermore, a study conducted in animal models demonstrated that angiotensin-converting enzyme (ACE)-2 (ACE2) plays an important role in reducing OS by converting angiotensin II into angiotensin, with a consequent attenuation of inflammatory signaling cascades [20]. Coronavirus 2 is able to bind cell membrane angiotensin-converting enzyme (ACE)-2 receptor [21], with the consequent release of the viral load [22]. The reduction in bioavailable ACE2 induces an overexpression of angiotensin II, that is able to activate the Nicotinamide Adenine Dinucleotide PHosphate (NADPH) oxidase pattern, leading to OS and inflammatory response [21,23]. During Severe Acute Respiratory Syndrome Coronavirus 2 (SARS-CoV-2) infection, ROS and RNS are able to regulate the gene expression via Nuclear Factor kappa B (NF-kB); when hyper-expressed, this transcription factor correlates to a progression of severe cases of Coronavirus disease-19 (COVID-19) [3,7]. The Nrf2 is also involved in SARS-CoV-2 infection; in particular, its higher levels are associated with a less severe form of COVID-19 [3]. Antioxidants capable of activating Nrf2 (i.e., dimethyl fumarate, BG12) could be used in the treatment of COVID-19 [24]. BG12 is a molecule able both to activate Nrf2 antioxidant response pathway and to inhibit the proinflammatory cascade. In particular, the efficacy of BG-12 has been demonstrated in clinical experiments involving patients with relapsing–remitting multiple sclerosis. In fact, the oral use of BG12, if compared to placebo, reduced the proportion of patients who had a clinical and radiological relapse [25]. A recent prospective study including 40 pediatric patients with a confirmed diagnosis of SARS-CoV-2 infection has been conducted by Gumus et al. to determine the relationship between Nrf2 and oxidative balance in this population. The study showed a decreased level of Nrf2 that was correlated to a decreased total antioxidant status, with a contextual increase in total oxidant status (TOS) and OS index (OSI) compared to the control group. The decrease in Nrf2 levels, through an increase in ROS production, might explain the lung tissue damage related to COVID-19; moreover, the more symptomatic the child was, the higher the TOS and OSI values were [26]. In particular, reduced Nrf2 levels were associated with bronchopulmonary inflammation, epithelial damage and mucous cell metaplasia [26]. An antioxidant diet, rich in vegetables (i.e., broccoli), can have a clinical benefit, although in clinical cases of SARS-CoV-2 affected patients who were given broccoli capsules, the clinical benefit was partial [24]. In addition, TRPA1 and TRPV1 play an important role in causing several symptoms of COVID-19 since they are able to mediate airway tissue injury and inflammation [24]. For this reason, another mechanism that could be used to reduce inflammation in SARS-CoV-2 infection is the activation of TRPA1/TRPV1 by dietary antioxidants. For example, TRPV1 localized in the sensory neurons can be desensitized by capsaicin (contained in red peppers); this consequent dose-dependent desensitization may be effective within minutes and for up to a few hours [24]. Aykac et al. [2] showed that serum native thiol and total thiol levels were significantly lower in adults and in children affected by COVID-19 than in controls (*p*-value = 0.001). Thiols are the main elements of the antioxidant defense system and a good indicator of cellular redox status [2]. In a study involving 160 hospitalized COVID-19 adult patients, Ducastel et al. demonstrated a reduction in thiol plasma concentration correlated with the severity of the disease [27]. These results support the hypothesis that thiol concentration could be used as a predictor of Intensive Care Unit admission (*p*-value < 0.001) [2,27]. Furthermore, thiol-derived compounds might be used to control the inflammatory response in COVID-19 patients. For instance, Disulfiram [28], a thiol-reacting Food and Drug Administration (FDA)-approved drug to treat alcohol use disorder, is a potent anti-inflammatory agent able to inhibit MERS-CoV and SARS-CoV proteases [2,28]. Finally, elevated interleukin (IL)-6 and calprotectin levels have been associated with an increased mortality among patients with SARS-CoV-2 infection [27]. Moreover, in a recent observational study conducted by Passoss et al., it has been demonstrated that the new coronavirus 2 is able to increase OS biomarkers’ serum levels, such as the glial fibrillary acidic protein (GFAP) and the receptor for advanced glycation end products (RAGE), whose increase is directly proportional to the level of neurological damage in patients affected by COVID-19 [23]. The main OS patterns in SARS-CoV-2 infection are included in Table 1.

Several agents endowed with antioxidant properties have been assayed or proposed to lower the risk of being affected by SARS-CoV-2 infection and/or to be used as an adjunctive treatment in case of severe COVID-19 [29]. Due to the encouraging preclinical data, we analyzed here the antioxidants on which more studies are available about the mechanisms of interactions with coronavirus 2 (Figure 1) and on which more clinical data are present in literature. These compounds seem to have a relevant role in modulating the immune response during respiratory viral infections (Table A1); starting from evidence on adults, we speculated on their potential use in children.

## 2. Methods

An electronic literature search was conducted using the Pubmed and Google Scholar databases. The following medical subject heading (MeSH) words were used individually in the search: antioxidants, vitamin A, ascorbic acid, flavonoids, hesperidin, quercetin, lactoferrin, resveratrol, zinc, selenium, melatonin and respiratory tract infection. In the latter case, the subheading “virology” was included. The following MeSH terms were used in combination: respiratory tract infections and antioxidants, respiratory tract infections and flavonoids, respiratory tract infections and quercetin, respiratory tract infections and hesperidin, respiratory tract infections and lactoferrin, respiratory tract infections and melatonin, respiratory tract infections and resveratrol, respiratory tract infections and vitamin A, respiratory tract infections and ascorbic acid, respiratory tract infections and zinc, respiratory tract infections and selenium, COVID-19 and antioxidants, COVID-19 and flavonoids, COVID-19 and quercetin, COVID-19 and hesperidin, COVID-19 and lactoferrin, COVID-19 and melatonin, COVID-19 and resveratrol, COVID-19 and vitamin A, COVID-19 and ascorbic acid, COVID-19 and zinc, and COVID-19 and selenium. The search was limited to studies published within the last decade (2011–2022) and selected based on the following eligibility criteria.

Exclusion criteria:Studies without full text available.Published studies in local languages, except for English.Non-relevant studies about other antioxidants, for the paucity of data in respiratory tract infections (i.e., copper, vitamin E, pentoxifylline).Studies about molecules that do not have a primary antioxidant role (i.e., vitamin D).Studies about the role of antioxidants in bacterial respiratory tract infections.Commentaries, letters and case-reports.

Inclusion criteria:In vitro studies about the interaction mechanism between coronavirus 2 and antioxidants.Clinical studies evaluating the potential role in preventing and/or treating SARS-CoV-2 infection of the main reviewed antioxidants in adults and children.

The reference lists of the retrieved articles were also consulted.

## 3. Role of Antioxidants in Prevention/Treatment of Coronavirus-19 Disease: Preclinical and Clinical Studies

### 3.1. Flavonoids

Flavonoids represent a large family of chemical compounds produced by plants under stressful conditions, with antioxidant, anti-inflammatory, immunomodulatory, antiviral and anti-aggregant properties [30]. There are many preclinical studies describing the antiviral activity of flavonoids, suggesting their potential use in SARS-CoV-2 infection [31,32].

Margolin et al. conducted a randomized clinical trial involving more than 100 subjects (aged 30 years and older) to evaluate the efficacy of an orally administered formulation containing zinc, vitamins and quercetin and found a significant difference in the occurrence of flu-like symptoms and SARS-CoV-2 infection over a 20-week observation period. Flu-like symptoms occurred less frequently in the supplemented group (4% vs. 20%), and none of the treated subjects developed SARS-CoV-2 infection compared with the untreated group, in which 75% of symptomatic subjects contracted the infection [33].

Another very interesting observation was made by Shawan et al., who identified luteolin as a possible ACE-2 inhibitor with a mechanism similar to that of the Food and Drug Administration (FDA)-approved drug remdesivir [34,35,36]. In a randomized, double-blind, case-control study in non-COVID-19 children aged 6 to 8 years, the efficacy of a mango juice by-product (JBP) in preventing gastroenteritis and upper respiratory tract infections (URTIs) was evaluated; the choice of mango stems from the fact that it contains polyphenols, including flavonoids. A significant reduction in respiratory symptoms was observed in the group of children who were offered mango-JBP at a dose of 2 g/day for two months (*p*-value ≤ 0.038), suggesting the immunomodulatory, antimicrobial and antiviral role of polyphenols contained in the administered product [37]. Starting from this finding, similar results could be assumed in children with URTIs by SARS-CoV-2, but clinical trials are currently unavailable in the pediatric population.

### 3.2. Quercetin

Quercetin is a polyphenolic compound belonging to the flavonoids whose antiviral and viricidal activity has been hypothesized in literature [31,32]. Chen et al. observed that an aqueous extract of *Houttuynia cordata* (HC) inhibits the replication of Herpes Simplex Virus (HSV) and SARS virus by means of a mechanism that inhibits the activation of NF-κB, which is necessary for viral replication [38]. Quercetin also inhibits the replication in cell cultures of Human Influenza A strains H1N1 and H_3_N_2_ as well as the entry of H5N1 [32]. It has been found that quercetin has an antiviral activity against SARS-CoV-1, which has a genome that is 79% comparable to that of SARS-CoV-2 [39,40]. Zhang et al. recently demonstrated the ability of quercetin to inhibit SARS-CoV-2 proteins 3CLpro and PLpro [41]. Moreover, quercetin plays a role in the modulation of the acid sphingomyelinase/ceramide system, which is involved in the internalization of SARS-CoV-2 into respiratory epithelial cells [42]. In a randomized clinical trial, Shohan et al. evaluated the therapeutic efficacy of quercetin administered in combination with antiviral drugs such as remdesivir and favipiravir in 60 adult patients with severe COVID-19. Patients in the control group received remdesivir or favipiravir, while, in the intervention group, quercetin (500 mg orally twice daily for seven days) was administered in addition to antiviral therapy. The only significant difference was found in the days of hospitalization following completion of therapy, which were lower in the intervention group than in the control (4.63 vs. 3.13 days). It should equally be reported that the differences were also close to statistical significance in the number of days of ICU hospitalization (8–15 vs. 6–10 days), number of deaths (3 vs. 0) and number of patients discharged (27 vs. 30), with a benefit in patients supplemented with quercetin [43].

In a randomized, controlled, open-label clinical trial by Di Pierro et al. conducted on 42 patients with mild COVID-19 (aged > 18 years) and treated at home, standard care (SC), consisting of administration of antipyretics and azithromycin, was compared with the administration of oral quercetin (600 mg three times daily on days 1 to 7 and 400 mg twice daily on days 8 to 14) added to conventional therapy. Specifically, the number of days necessary for the negativization of the RT-PCR for SARS-CoV-2 and the improvement in symptoms was evaluated. A faster negativization was observed in the quercetin-treated group than in the SC group (76% vs. 9.5% negative on day 7). Moreover, in patients treated with quercetin, the absence of symptoms was observed on day 7 in 57% of cases compared to 19% in those treated with SC [44]. There are no clinical studies conducted in children evaluating the benefits of quercetin administration in the prevention and/or therapy of SARS-CoV-2 infection. Thus, to date, there is no scientific evidence supporting the administration of quercetin in the pediatric population, but the available evidence in adults suggests the need for further studies.

### 3.3. Hesperidin

Hesperidin is a common flavone glycoside found in citrus fruits with anti-atherogenic, anti-hyperlipidemic, antidiabetic, venotonic, cardioprotective, anti-inflammatory and antihypertensive properties. Hesperidin seems to interfere with the replication of SARS-CoV-2 and to prevent the virus’ entry into cells by acting at the level of the ACE-2 receptor and the spike protein. In addition, hesperidin would activate the interferon-mytogen-activated protein kinase pathway that reduces viral replication; finally, there would be a rationale in its use in therapy, which derives from its ability to inhibit the secretion of proinflammatory cytokines, responsible for the symptoms of the disease, up to the most severe forms with Acute Respiratory Distress Syndrome (ARDS) [45]. Since SARS-CoV-2 infection is associated with an increased risk of venous thromboembolism, co-administration of low-molecular-weight heparin (LMWH) and a mixture of flavonoids, including hesperidin and diosmin, would appear to be favorable in preventing deep vein and pulmonary thrombosis, as demonstrated by a randomized, single-blind, placebo-controlled, cross-over study of adult subjects with increased cardiovascular risk in which endothelial function, measured as flow-mediated dilation, improved significantly (5.7% vs. 7.9%) in the group of subjects given 500 mL/day of orange juice [45,46,47].

### 3.4. Lactoferrin

Lactoferrin is a multifunctional and pleiotropic protein present in several biological secretions, such as human milk, with higher concentrations in the *colostrum* (up to 7 g/L) [48,49]. It exerts a strong antiviral activity [48] by both stimulating the immune system and limiting the inflammatory response of the host [50]. Moreover, this protein can enter the host cells and translocate into the nucleus, influencing pro-inflammatory gene expression [51]. It has been demonstrated that lactoferrin production increases during common coronavirus infection [52]. In particular, during the first 2003 SARS-CoV epidemic, researchers had already studied the role of this protein, showing its ability to negatively influence viral replication by enhancing Natural Killer (NK) cell activity and stimulating neutrophil aggregation and adhesion [48]. Lactoferrin is able to exert its antiviral activity either by direct attachment to the viral particles or by obscuring their cellular receptors [53]. In preclinical studies in 2011, Lang et al. [48] had already demonstrated that lactoferrin was able to interact with heparan sulfate proteoglycans (HSPG), located on the cell surface, and, today, recognized as a binding site for coronavirus 2 [53]. For this reason, oral supplementation of lactoferrin might be used to prevent COVID-19 by avoiding the initial interaction between coronavirus 2 and host cells, and some speculative data are also available for its potential therapeutic use [48,52]. In fact, Serrano et al. [49] have shown, in a prospective observational study conducted in 75 adults affected by COVID-19, that the use of liposomal bovine lactoferrin as a food supplement was correlated with an improvement in symptoms during the first four days of treatment (−22% of patients with dry cough; −44% of patients experiencing muscle pain; −66% of patients reporting weakness; none reporting headache). Therapeutic dose was 64–96 mg every 6 h daily, whereas individuals in contact with symptomatic patients were treated with half of the curative dose (64 mg two or three times a day), resulting in disease prevention [49]. In particular, adult patients presenting with dry cough and nasal congestion were also treated with liposomal lactoferrin nasal drops, demonstrating an improvement in respiratory symptoms [49]. These findings are supported by another recent randomized, prospective, interventional pilot study conducted by Algahtani et al., involving 54 adults with SARS-CoV-2 confirmed infection. Patients were divided into three groups and were randomized to not receive any oral supplementation, receiving 200 mg of lactoferrin once a day and receiving 200 mg of lactoferrin twice a day. After seven days, there was an improvement in symptoms (i.e., fever, dry cough and headache) in both groups of treated patients, although the difference was not statistically significant compared to the control group [54]. However, no published data are available about its clinical use in the pediatric population.

### 3.5. Melatonin

Melatonin is a hormone that plays an important role in immune function and metabolism of the host [55]. In vitro studies have already demonstrated that its use can increase antiviral drugs’ power by inhibiting the replication of the influenza virus and RSV [56]. Starting from these assumptions, several authors have supposed that an oral administration of melatonin at high doses could also have both a prophylactic and therapeutic role in SARS-CoV-2 infection [55]. Indeed, Köken et al. [57], in a cross-sectional study on 84 patients (7–15 years and older), showed lower concentrations of melatonin in children affected by SARS-CoV-2 infection compared to the control group, though the difference was not statistically significant, except for the COVID-19 patients in the 7–12 age group [57]. Farnoosh et al. [58] recently demonstrated the efficacy of melatonin’s oral use as an adjuvant therapy in mild to moderate COVID-19 adult patients. In particular, those treated for two weeks with 3 mg of melatonin, three times a day, showed a significant improvement in respiratory symptoms (cough, dyspnea) and fatigue, as well as a significant reduction in C-reactive protein serum levels and an improvement in the chest X-ray. Many anti-inflammatory mechanisms of melatonin are known: it inhibits the expression of inflammasome genes, the inflammasome activation and the translocation and expression of NF-kB [55]. All of these inflammatory patterns underlie the pathogenetic mechanism of the new coronavirus 2 [56], so melatonin might positively influence the immune regulation during this viral infection [55]. In particular, it could interfere with blood coagulation and vascular inflammation, reducing the endothelium damage during the cytokines’ storm in COVID-19, which is also achieved by facilitating mitochondrial oxidative phosphorylation [55]. According to this, a randomized clinical trial conducted by Hasan et al. [59] showed that a dose of 10 mg oral melatonin added to SC was more effective than SC alone in severe COVID-19 patients. In particular, a significant reduction in thrombosis events, sepsis onset and mortality rate was observed in the treated group vs. controls (*p*-value < 0.05) [60]. An ongoing randomized multicenter clinical trial is evaluating the efficacy of melatonin in the prophylaxis of SARS-CoV-2 infection in high-risk healthcare workers: doses between 5 and 10 mg (depending on age) could be useful [55], although the safety of this hormone is not completely known [56]. Finally, it has been supposed that melatonin could be used as an adjuvant in the vaccination strategy, especially in children, due to its immunomodulating properties [56], but further research is needed.

### 3.6. Zinc and Selenium

Zinc and selenium play crucial immunomodulatory functions and are able to influence the course of several viral infections. Indeed, these trace elements may prevent the entry of viruses into the host cells and limit viral replication [61].

Zinc must be introduced by the diet, as the organism is unable to produce or store it and its deficiency is correlated to immune system dysfunction [62]. This oligoelement is able to increase interferon production, by modulating inflammatory cytokines; this action may explain its role in reducing the incidence of acute respiratory infections in a group of 301 orally supplemented children (2–5 years of age) compared to a placebo group [63,64]. It has been hypothesized that zinc could be useful as an adjuvant therapy in combating COVID-19, given its potential to inhibit Coronavirus RiboNucleic Acid (RNA) polymerase activity [65]. Moreover, it has been shown that zinc is able to interfere with ACE-2 activity, which is required for SARS-CoV-2’s entry into the host cells [61]. For these reasons, a randomized controlled study was conducted by Abd-Elsalam et al. to evaluate whether oral zinc supplementation, associated with chloroquine/hydroxychloroquine therapy, could be useful in treating COVID-19 adults. No statistically significant results were found [66]. Nonetheless, the synergistic effect of zinc and antiviral treatment has also been tested, providing good results, as were demonstrated in a phase I clinical trial [67]. In particular, data from this study conducted in adult patients have shown that the combined use of nitazoxanide, ribavirin and ivermectin plus zinc supplement effectively cleared SARS-CoV-2 from the nasopharynx in a shorter time than supportive symptomatic treatment alone [67]. Moreover, a randomized clinical trial conducted in adults affected by idiopathic dysgeusia has shown a beneficial role of oral zinc supplementation in improving gustatory sensitivity [65]. Additionally, in a recent prospective clinical trial, COVID-19 patients treated with an oral zinc supplementation (220 mg zinc sulfate equivalent to 50 mg elemental zinc twice daily) had a lower duration of anosmia/hyposmia than those who did not receive zinc therapy [68].

Selenium is another essential trace element in humans: most of its benefits are due to its incorporation into selenocysteine, a component of selenoproteins that have an important antioxidant role [69]. There is evidence in adults showing the correlation between serum selenium deficiency and the increase in cellular oxidative stress during influenza and Coxsackie virus infections [69]. A recent cross-sectional study conducted by Majeed et al. in 30 adults affected by SARS-CoV-2 infection, demonstrated that selenium serum levels were significantly lower compared to the control group (*p*-value = 0.0003) [70,71]. Previous studies conducted in vitro have already demonstrated the fundamental role of selenium in reducing OS, cellular apoptosis and, also, platelet aggregation. Therefore, it is likely that selenium might be useful in COVID-19 patients by influencing the pathogenetic mechanism underlying the infection [61]. In animal models, it has been shown that the deficiency of this trace element could increase coronavirus’ pathogenicity, because of increasing mutation of the viral genome [61]. In a recent pilot double-blind placebo-controlled randomized clinical trial, selenium parenteral supplementation as sodium selenite in adults with ARDS improved respiratory mechanics of the treated patients [69,72]. Nonetheless, further studies are necessary to evaluate selenium oral supplementation as adjuvant therapy or prophylaxis in SARS-CoV-2 infection, especially in children.

### 3.7. Vitamins

Several studies have shown that ascorbic acid (AA) has effects on the immune system and may increase antimicrobial activity; this would suggest its potential role in the prevention and treatment of infections, especially those affecting the respiratory tract [73,74,75]. Although the biomolecular mechanisms are not clearly defined, it seems that AA acts on the cells of the immune system by inducing epigenetic modifications; for example, ten-eleven translocation proteins (TETs) and Jumonij-C domain-containing histone demethylases (JHDMs), which regulate the transcription of gene encoding factors involved in the innate and adaptive immune response, are activated by the reduction of Fe^3+^ to its catalytically active form (Fe^2+^) by an oxidation–reduction mechanism in which ascorbate acts as an electron donor [73]. AA is most likely to promote proliferation and differentiation of T-helper 1 (Th-1) and Th-17 lymphocytes in a dose-dependent way, chemotaxis and phagocytosis mechanisms of granulocytes, and monocyte–macrophage cells and caspase-mediated apoptosis of leukocytes, reducing inflammatory processes and promoting regeneration of involved tissues (i.e., respiratory epithelium) [76]. There are still unclear findings about the role of AA in the differentiation of cytotoxic T lymphocytes, B lymphocytes and NK cells. AA seems to be able to reduce the production of pro-inflammatory cytokines such as tumor necrosis factor (TNF)-α and IL-6. Finally, AA could also interact with mediators involved in the dysreactive condition that is determined during sepsis, such as the Epidermal Growth Factor Receptor (EGFR), Mitogen-Activated Protein Kinase-1 (MAPK1), Proto-oncogene c (JUN), C-C chemokine Receptor type 5 (CCR5), Mitogen Activated Protein Kinase 3 (MAPK3), Angiotensin II receptor type 2 (AGTR2) and Signal transducer and activator of transcription-3 (STAT3) [73].

The most debated issues include the route of administration (enteral or parenteral), dosage (low or high) and the appropriate time to administer AA (as prophylaxis or as therapy).

In a meta-analysis by Gomez et al., it was found that enteral administration of 80 mg/day of AA did not prevent cold in healthy adults and children [77]. In a Cochrane systematic review, it was shown that regular AA supplementation (1 to 2 g/day) in children reduced cold duration by 14% and symptom severity, but it was not effective in preventing infections [63]. In a meta-analysis by Ran et al. that examined nine randomized controlled trials on subjects of all ages, it was found that daily enteral administration of low-dose AA (maximum dose 1 g/day) and supplementation at the time of onset of cold symptoms (up to 3–4 g/day) were associated with a significant shortening of disease duration compared with the control group (MD = −0.56, 95% CI [−1.03, −0.10] [78].

There are not yet available studies on the use of AA in the prevention and treatment of SARS-CoV-2 infection in children, but relevant findings were collected in adult patients. In a randomized double-blind clinical trial conducted by Majidi et al., 120 patients (age range 35–75 years) with severe COVID-19 requiring intensive support were enrolled to evaluate enteral ascorbic acid administration (500 mg daily for 14 days). At day 14, a significant difference in survival was observed in the supplemented group (16.1% vs. 2.9%) [79]. In a two-center retrospective study conducted by Al Sulaiman et al. including 739 critically ill patients admitted to the ICU with confirmed COVID-19, 158 of whom had been given ascorbic acid (1000 mg/day for once daily) enterally, the supplemented group showed a significant decrease in the incidence of thrombosis (6.1% vs. 13% OR 0.42 95% CI [0.184–0.937]) [80]. Based on the assumption that high doses of intravenous vitamin C would be considered effective in the treatment of bacterial sepsis and ARDS [81], a randomized controlled clinical trial was conducted on 56 patients aged 18–80 years with severe COVID-19 admitted to the ICU who were given high-dose intravenous vitamin C (24 g/day) or placebo for 7 days. At the end of the observation period, there was no significant difference either on days free from invasive mechanical ventilation or 28-day mortality; however, an improvement in oxygenation was observed in vitamin C-treated patients, recording a significant increase in PaO_2_/FiO_2_ at day 7 (229 vs. 151, 95% CI 33–122) as well as a lower mean value of IL-6 (19.42 vs. 158, 95% CI −301.72 to −29.79) [82]. Finally, in an open-label randomized controlled clinical trial involving 214 patients with SARS-CoV-2 infection randomized to receive only conventional therapy or, in addition, either zinc gluconate (50 mg/day) or ascorbic acid (8000 mg/day) or both for 10 days, no significant difference was observed in the number of days needed to achieve a 50 percent reduction in symptoms, as measured by a special score based on the assessment of fever, cough, fatigue and dyspnea (6.7 vs. 5.5. vs. 5.9 vs. 5.5 in the group receiving standard of care, ascorbic acid, zinc gluconate or both, respectively) [83]. Based on this evidence, we could speculate that AA would be useful in the prevention and treatment of respiratory infections, including SARS-CoV-2. Nonetheless, further studies are needed, especially in children.

Vitamin A (VA) and retinoic acid (RA) are involved in the regulation of the immune system; in particular, vitamin A deficiency (VAD), even in its subclinical form, results in impaired T and B lymphocyte function and dysregulates the innate immune response mediated by NK cells, monocytes, dendritic cells and cytokines, such as type I interferon [84,85]. In an animal model studied by McGill et al., the effect of VAD was evaluated on calves given an intranasal vaccine against Bovine Respiratory Syncytial Virus (BRSV), showing that calves with VAD did not immunize after vaccination and had a more severe course of respiratory infection than calves with normal VA stores. Moreover, in the same model, it was observed that the effect of acute respiratory viral infection results in depletion of VA stores in the liver. This could suggest the usefulness of VA supplementation in convalescent subjects to avoid long-term VAD and susceptibility to secondary infections or complicated sequelae, although these findings cannot be directly translated to humans [86]. VA is known to promote epithelial regeneration, tissue repair and healing after an infectious event, and previous studies have speculated that it may interfere with the course of respiratory infections, although the results are unclear [87]. In a case-control study that included children aged 4 to 6 years, a significantly lower mean serum retinol level was observed in the group of those with respiratory infections compared with the group of healthy controls (32.4 ± 13.12 mg/dL vs. 56.9 ± 29.82 mg/dL). In the same study, an association was found between hepatic retinol stores and recovery of respiratory function after infection; specifically, by the assessment of respiratory function, earlier recovery was observed in children with URTIs and greater deposits of VA in the liver [88]. The underlying biochemical mechanisms are unclear, but RA is known to promote the maintenance of regulatory T-cells’ (T-reg) lymphocytes that reduce airway inflammation. In a study on Ecuadorian children, a low dietary intake of VA during the first five years of life was associated with a depletion of T-reg lymphocytes and an increased risk of developing bronchial hyperreactivity [89,90]. To date, there are no trials evaluating the efficacy of VA supplementation in the prevention and treatment of SARS-CoV-2 infection in children, but there are preclinical findings that suggest the benefits of supplementation and coadministration with vaccines [91]; based on these findings, clinical trials should be conducted in the pediatric population, with the aim of investigating the usefulness of VA supplementation and its potential clinical indication.

### 3.8. Resveratrol

Resveratrol is a natural non-flavonoid polyphenol that can be found in fruits and vegetables, with a potential anti-infective and anti-inflammatory role [92].

Preclinical studies, conducted in Vero cells, have demonstrated that resveratrol can inhibit the replication of SARS-CoV-2, especially after the virus adsorption step, up to 40 h post-infection. At the same time, if resveratrol is present during the viral adsorption phase, the molecule can interfere with viral entry into cells by reducing viral RNA synthesis by more than 60%. Thus, resveratrol might influence the infectious cycle at an early stage, before the virus’ assembly [93]. A double-blind, placebo-controlled cross-over trial conducted by De Ligt et al. involving 11 otherwise healthy obese males, randomized to 30 days of placebo or 30 days of 150 mg/day resveratrol, demonstrated that this supplementation significantly reduced the expression of ACE2 in human adipose tissue (AT) (*p*-value = 0.029). Moreover, resveratrol supplementation reduced AT leptin expression, with a consequential potential benefit on COVID-19 affected patients. In fact, higher levels of serum leptin have been correlated to more severe forms of the disease, especially in obese individuals [22]. Nevertheless, it is difficult to use resveratrol alone in clinical practice, because of its poor stability in aqueous solutions and the low bioavailability: indeed, after oral administration, the molecule is rapidly and extensively metabolized [94]. For this reason, an inhaled formulation that combines resveratrol and carboxymethylated glucan (CM-glucan) has been proposed, which is able to stabilize and enhance resveratrol’s biological activities. The aim is to control the early stages of the infection, avoiding viral spread to the lower airways [94]. Varricchio et al. [92] had already demonstrated that in 82 children (mean age 8.1 +/− 2.6 years) with Respiratory Recurrent Infections (RRI), resveratrol plus CM-β-glucan, in an aerosolized solution, could reduce nasal symptoms, cough and fever compared to a control group. More recently, Baldassarre et al. [95], in a double-blind, randomized, placebo-controlled clinical trial, demonstrated that in a group of 89 infants from 0 to 6 months of age, the use of a nasal solution containing resveratrol plus CM-β-glucan may have a potential therapeutic role in reducing respiratory symptoms (cough and sneezing) and relapses in children affected by the common cold. Clinical studies investigating the possible use of resveratrol in prophylaxis and/or treatment of SARS-CoV-2 infection in adults and in children are still needed.

## 4. Conclusions and Future Perspectives

The uncontrolled spread of SARS-CoV-2 is a health and social issue that has plagued the world for more than two years. However, today, we can take advantage of prevention and treatment measures that are based on passive and active immunization mechanisms. The administration of multiple doses of messenger-RNA vaccine together with hygienic norms (especially mask use and physical distancing) have determined an epidemiological change and the prevalence of less severe clinical forms of COVID-19, rarely requiring intensive support and treatment. The topic of this narrative review stems from the interest in nutrition and the constant questioning of the role that diet could play in the prevention and treatment of various diseases, including viral respiratory infections. It has been speculated that a dietary supplementation may be of high importance in the current COVID-19 pandemic, so several studies on this topic have been initiated [70]. Nevertheless, this issue is still highly controversial, particularly in relation to SARS-CoV-2 infection in the pediatric age group. We found that antioxidants might have a protective and therapeutic role against COVID-19 and, starting from clinical data obtained in adults (Table 2), this review provides an opportunity to consider and speculate on whether new trials in children should be conducted, allowing for stronger scientific evidence than those currently available. The benefit that could be obtained would be significant, since most of these substances are included in a free non-selective diet and are easily available. However, more studies are necessary to evaluate the real safety and efficacy of these compounds; an additional issue to be investigated concerns bioavailability, and therefore, data comparing the effects obtained according to different routes of administration are also necessary. In conclusion, although COVID-19 in children is generally a mild disease, the availability of compounds such as antioxidants which could be able to prevent SARS-CoV-2 infection and/or reduce the length of respiratory symptoms could presumably have significant epidemiological effects, reducing both medical burden and social cost.

## Figures and Tables

**Figure 1 antioxidants-11-01638-f001:**
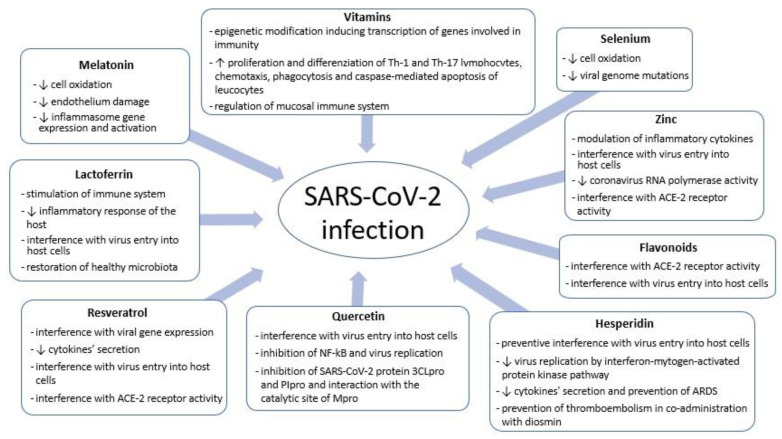
SARS-CoV-2 and antioxidants: mechanisms of interaction between coronavirus 2 and host cells.

**Table 1 antioxidants-11-01638-t001:** Oxidative stress patterns involved in SARS-CoV-2 infection.

Oxidative Stress Pattern	Mechanism of Action	SARS-CoV-2 Infection
**ACE2**	It converts angiotensin II into angiotensin, with a consequent attenuation of inflammatory signaling cascades	The reduction in bioavailable ACE2 induces an overexpression of angiotensin II, which is able to activate OS patterns and inflammatory response
**Nrf2**	It contributes to the expression of cell-protective genes in response to OS	The decrease in Nrf2 levels, through an increase in ROS production, might explain airway inflammation and tissue damage
**TRPA1/TRPV1**	They are sensors of OS and are able to mediate airway inflammation and tissue injury	COVID-19 morbidity may be associated with TRPA1 and/or TRPV1 activation as well as desensitization
**Thiols**	They have antioxidant and anti-inflammatory properties	Reduced thiol plasma concentration has been correlated with severity of COVID19

ACE2: angiotensin-converting enzyme-2; Nrf2: Nuclear Erythroid Factor 2-Related Factor 2; ROS: Reactive Oxygen Species; TRPA1/TRPV1: Transient Receptor Potential Ankyrin 1/Transient Receptor Potential Vanilloid 1.

**Table 2 antioxidants-11-01638-t002:** Potential clinical use of antioxidants in prophylaxis and/or treatment of SARS-CoV-2 infection in adults.

Type of Antioxidant	References	Country	Study Type	Study Population	Aim of the Study	Results
**Quercetin**	Shohan et al. [43]	Iran	Randomized controlled clinical trial	60 adults (age > 18 years) affected by severe forms of COVID-19 who were not admitted to the ICU	Evaluating the therapeutic efficacy of quercetin administered in combination with antiviral drugs such as remdesivir and favipiravir in COVID-19 adult patients	Daily enteral administration of quercetin in addition to antiviral therapy results in a significant reduction in days of hospitalization following completion of therapy. Results close to statistical significance in the number of ICU admission days, number of deaths and number of discharges with benefit in the group of patients supplemented with quercetin.
	Di Pierro et al. [44]	Italy	Randomized controlled open-label clinical trial	42 patients affected by mild COVID-19 (age > 18 years) treated at home	Compare standard care with oral administration of quercetin in addition to conventional therapy in COVID-19 adult patients	Faster SARS-CoV-2 RT-PCR negativization and improvement of symptoms in patients supplemented with oral quercetin.
**Lactoferrin**	Serrano et al. [49]	Spain	Prospective observational study	75 adults (median age 42 years) affected by COVID-19	Evaluating the therapeutic role of lactoferrin in SARS-CoV-2 infection and its preventative role in individuals in contact with symptomatic patients	The oral liposomal bovine lactoferrin administration at a therapeutic dose (64–96 mg every 6 h daily) in the first 5 days of infection caused improvement in respiratory symptoms (headache, muscular pain, taste, smell, weakness and dry cough) and a faster recovery compared to a control group. Individuals in contact with symptomatic patients, treated with half of the therapeutic dose, had a benefit in disease prevention.
	Algahtani et al. [54]	Egypt	Randomized prospective interventional pilot study	54 adults (median age 48 years) affected by COVID-19	Evaluating the therapeutic role of lactoferrin in SARS-CoV-2 mild to moderate infection	Oral supplementation of lactoferrin is associated with an improvement of symptoms (fever, dry cough, diarrhea, headache, loss of sense of taste and/or smell and tiredness) after 7 days of lactoferrin treatment, although not statistically significant. At the same time, the improvement in laboratory markers (hemoglobin and albumin increase; liver enzymes, lactate dehydrogenase and C-reactive protein reduction) was not statistically significant in the treated groups compared to the control one.
**Melatonin**	Farnoosh et al. [58]	Iran	Single-center double blind randomized clinical trial	74 patients (>18 years old) affected by mild to moderate COVID-19	Evaluating the efficacy and safety of oral melatonin in combination with standard treatment in adult hospitalized patients affected by COVID-19	The administration of oral melatonin (3 mg, three times a day) combined with standard of care was correlated to an improvement in respiratory symptoms (cough, dyspnea) and fatigue. Moreover, in treated patients, there was an improvement also in laboratory (C-reactive protein serum levels) and radiologic (chest X-ray) exams. Finally, return to baseline health was significantly shorter in the patients receiving melatonin supplementation.
	Hasan et al. [59]	Iraq	Randomized clinical trial	158 patients (>18 and <80 years old) affected by mild to moderate COVID-19	Evaluating the efficacy of oral melatonin in combination with standard treatment in adult hospitalized patients affected by COVID-19	The administration of oral melatonin (10 mg/day), associated with standard of care, was more effective than standard of care alone in affected severe COVID-19 patients. In particular, it exerted a potential role in reduction of thrombosis events, sepsis onset and mortality rate in the treated group.
	Köken Yayici O et al. [57]	Turkey	Cross-sectional study	84 patients (7–15 years and older) affected by COVID-19	Evaluating serum melatonin concentration in children affected by SARS-CoV-2 infection	The study showed a lower concentration of melatonin in SARS-CoV-2 affected children in the 7–12 age group; no statistical difference in the other age groups. No changes in sleep patterns in affected children.
**Zinc**	Abd-Elsalam et al. [66]	Egypt	Randomized controlled study	191 patients (median age 43 years) affected by COVID-19	Evaluating the efficacy of oral zinc supplementation in patients treated with chloroquine/hydroxychloroquine	Patients treated with oral zinc supplementation showed neither clinical nor laboratory improvements if compared to the control group (no statistically significant results).
	Elalfy et al. [67]	Egypt	Non randomized controlled trial	113 patients divided into two groups (<35 years and >35 years) affected by mild or moderate COVID-19	Evaluating the oral synergistic effect of zinc when associated to a triple therapy (nitazoxanide, ribavirin and ivermectin) in COVID-19 affected patients	The combination of nitazoxanide, ribavirin and ivermectin plus zinc was effective in suppressing the shedding of SARS-CoV-2 in nasopharyngeal swabs compared to those receiving routine supportive symptomatic treatment alone.
	Abdelmaksoud et al. [68]	Egypt	Prospective clinical trial	134 patients (median age 52 years) affected by mild to extremely severe COVID-19	Evaluating how many patients had smell or taste disorders and, among these, how oral zinc supplementation could interfere with the median duration of complete recovery	The median duration of taste and/or smell recovery was significantly shorter among patients who received zinc therapy (220 mg zinc sulfate, equivalent to 50 mg elemental zinc, twice daily) than those who did not receive zinc, while the median duration of complete recovery from COVID-19 was not significantly different among the two groups.
**Selenium**	Majeed et al. [71]	India	Cross-sectional study	30 adults (18–45 years of age) affected by COVID-19	Evaluating serum selenium concentration in adults affected by SARS-CoV-2 infection	The study showed a lower concentration of selenium in SARS-CoV-2 affected adults compared to the control group.
**Ascorbic acid**	Majidi et al. [79]	Iran	Randomized double-blind clinical trial	120 adults (age range 35–75 years) affected by severe forms of COVID-19 with intensive support needs	Evaluating enteral administration of 500 mg ascorbic acid daily for 14 days in adults affected by COVID-19	At day 14, a significant difference in survival was observed in favor of the ascorbic acid-supplemented patient group.
	Al Sulaiman et al. [80]	Saudi Arabia	Two-center, non-interventional retrospective cohort study	739 patients (≥18 years-old) affected by COVID-19, 158 of whom had been given ascorbic acid enterally	Evaluating enteral administration of ascorbic acid in COVID-19 adult patients	In the ascorbic acid-supplemented group there was no significant decrease in mortality, but there was a decrease in the incidence of thrombosis.
	Zhang J et al. [82]	China	Randomized controlled clinical trial	56 patients aged 18–80 years affected by severe COVID-19 admitted to the ICU	Compare high-dose intravenous vitamin C (24 g/day) with placebo for 7 days	At the end of the observation period, there was no significant difference either on days free from invasive mechanical ventilation or 28-day mortality; an improvement in oxygenation was observed in vitamin C-treated patients and, similarly, a lower mean value of IL-6.
	Thomas et al. [83]	United States	Open-label randomized controlled clinical trial	214 adult patients (18 years or older) affected by SARS-CoV-2 infection	Compare the administration of conventional therapy alone with the addition of either zinc gluconate or ascorbic acid or both for 10 days	Nonsignificant difference in the number of days needed to achieve a 50 percent reduction in symptoms.
**Resveratrol**	De Ligt et al. [22]	Netherlands	Randomized double-blind placebo controlled crossover trial	11 obese males (median age 53 years) not affected by COVID-19	Evaluating the effects of 30-days resveratrol supplementation on renin–angiotensin–system components in the adipose tissue of otherwise healthy obese men	Resveratrol supplementation reduces ACE2 expression in human adipose tissue so, the prophylactic use of resveratrol in obese individuals could make them less susceptible for SARS-CoV-2 infection. Moreover, resveratrol reduces leptin serum levels in this population and its use could be beneficial as supplementary therapy in COVID-19 severe forms.

COVID-19: Coronavirus disease-19; ICU: intensive care unit; SARS-CoV-2: Severe Acute Respiratory Syndrome Coronavirus 2; RT-PCR: real-time polymerase chain reaction; ACE2: angiotensin-converting enzyme 2; IL-6: interleukin 6.

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
