# Peer review of "Antioxidant Effects of Dietary Supplements on Adult COVID-19 Patients: Why Do We Not Also Use Them in Children?"

_antioxidants, 2022, doi:10.3390/antiox11091638_

Round 1

Reviewer 1 Report

This manuscript describes the usefulness of food nutrients known as antioxidants for viral respiratory infections and their potential for prevention and treatment of COVID-19 infections.

Major points:

This review might be helpful in understanding the antioxidant effects of nutrients in foods against viral respiratory infections. However, the authors showed the effects of antioxidants in vitro reports with SARS-CoV2 and some clinical trials in adult patients with COVID-19. To date, clinical trials for pediatric COVID-19 patients do not yet exist for all of these nutritional supplements; therefore, it is an exaggerated expression to state the potential efficacy for pediatric patients with COVID-19 by citing evidence of efficacy in viral respiratory infections other than SARS-CoV2. It may be desirable to configure the review as “antioxidant effects of dietary supplements on adult COVID-19 patients.” In addition, the evidence presented in this review is a mixture of in vitro studies and clinical trials in humans. Adding a table showing clinical trials of antioxidants against COVID-19 in adults, including negative data, will deepen the reader’s understanding.

Minor Point 1: The text size in Figure 1 is too small and needs to be changed larger.

Minor point 2: The p-value in the manuscript is not required and it is desirable to show only the results with significant differences in previous reports.

Overall, we propose a major revision for this manuscript.

Reviewer 2 Report

Comments to the Author

The manuscript “Use of Antioxidants in Respiratory Viral Infections: Starting Point for Prevention and Treatment of the New SARS-CoV-2 Pandemic, from Adults to Children”. By Veronica Notarbartolo, et.al.,. The authors review the use of antioxidants to prevent/treat disease severity of SARS-CoV-2 and respiratory virus. The manuscript is interesting and well written. However, some additional information can improve the paper. 

Mayor

The authors may want to expand the explanation of the mechanism involved in RTI attenuation with antioxidants, by starting with a description of the immunomodulatory mechanism of the antioxidants, before jumping to SARS-CoV-2. That can give a better background for the readers on the importance of these immunomodulators.

The authors could add more information on NRf2, TRPA1, and TRPv1

I think if the authors suggest using antioxidants as an immunomodulator, it would be helpful to mention compounds used for this besides food (broccoli), that it is not something that can be administrated in a dose-depended manner. The authors could mention an antioxidant diet, rich in vegetables, like broccoli, but not as the leading example for this. There is a compound name BG12 that can be used as an example. 

< Preliminary clinical studies seem to sustain this hypothesis, but further studies are needed>. Describe these studies and references. 

<Evidence suggests that during respiratory viral infections, the use of antioxidants may help reduce symptoms and facilitate recovery in affected children [13,14], especially if they are administered early before the beginning of a severe pro-inflammatory response in the host>

I would like to know the details of this. What kind of antioxidants? What sort of RVI? This is the exciting part that would back up the author's ideas. 

< Aykac et al. have shown that serum native thiol and total thiol levels were significantly lower in adults and children affected by COVID-19 than in the control group (p-value=0.001).> needs to explain the relevance of this finding. There is a lack of connection between the essential information and the significance of the study. The review needs to summarize the information for the reader and give the reader all the information required to follow the main idea of the review. Give some introductory sentences about thiol and their importance in your manuscript. 

Thiol-derived compounds: here, the authors could mention some of this and if any publication that has used them to control inflammatory responses can back up this idea. 

Adding a summary table can help to visualize all the information. 

Minor

I would use upregulation instead of NRF2 hyperexpression, but that is up to the authors.  

Reference: <For this reason, another mechanism that could be used to reduce inflammation in SARS-CoV-2 infection is the activation of TRPA1/TRPV1 by some antioxidant ingredients in food (i.e., capsaicin), with a consequent dose-dependent desensitization that may be effective within minutes and for up to a few hours>. 

< There are many preclinical studies describing the antiviral activity of flavonoids, suggesting their potential use in SARS-CoV-2 infection [18,19].> two are not many 

Reference 29,30. These are pre-print since 2020, and I would be careful in using this information since these manuscripts haven’t been peer-reviewed. 

Round 2

Reviewer 1 Report

The reviewer thinks that the manuscript has been revised appropriately.

This manuscript is a resubmission of an earlier submission. The following is a list of the peer review reports and author responses from that submission.

Round 1

Reviewer 1 Report

This work is interesting basically. The information presented by the authors will be helpful for exploring novel therapeutic strategies for severe SARS-CoV-2 infection. The work will be strengthened if the authors present some evidence suggesting the pathobiochemical roles of increased oxidative stress status in SARS-CoV-2 infection. The authors should thoroughly search for the recent clinical and experimental studies on this matter. Especially, human studies in which several oxidative stress biomarkers are evaluated in association with the clinical condition should be presented in the paper.

Reviewer 2 Report

Thank you very much for the opportunity of reviewing this paper. The topic of this review is  interesting. Hovewer, it has important limitations.

The manuscript is a narrative review and although the requirements for this type of work are not as high as for a systematic review, they still exist to assess the quality of the review.

So many relevant items are missing from this review, which makes the review low-quality The most important are:

  • Description of literature search is missing. It’s not necessary to describe the literature search as much detail as in systematic review, but it’s necessary to specify search terms, types of literature included, quantity and names of the databases searched. If you give more information about terms search it would be easier to understand why you analyzed only 9 antioxidants.
  • The title of the manuscript does not match. In the manuscript you analyzed some clinical trials that were also conducted in adult patients.
  • In section: “ Antioxidants in clinical use to prevent/treat Coronavirus-19 disease (COVID-19)” you analyzed not only clinical studies but mainly in vitro studies…
  • The manuscript does not explain why you chose antioxidants as remedies for Covid-19, which mechanism of these compounds plays an important role in antiviral activity. Why didn't you just analyze different natural compounds or nutraceuticals. Moreover, sometimes you write about antioxidants, sometimes about nutraceuticals or micronutrients, keeping in mind the same compounds.

My overall rating is very low. The manuscript is very badly organized. Some in vitro / in vivo studies and some clinical studies are cited without any deep thought.

It is impossible to draw any conclusions after reading it.